# Techno-Functional Properties of Corn Flour with Cowpea (*Vigna unguilata*) Powders Obtained by Extrusion

**DOI:** 10.3390/foods12020298

**Published:** 2023-01-08

**Authors:** Luz Indira Sotelo-Díaz, Marta Igual, Javier Martínez-Monzó, Purificación García-Segovia

**Affiliations:** 1Food Investigation, Process Management and Service Group, Food Science and Culture Department, Universidad de La Sabana, Campus del Puente del Común Km. 7, Autopista Norte de Bogotá, Chía 250001, Colombia; 2Food Investigation and Innovation Group, Food Technology Department, Universitat Politècnica de València, Camino de Vera s/n, 46022 Valencia, Spain

**Keywords:** extrusion, cowpea, physicochemical properties, protein

## Abstract

Legumes are a good source of vegetal protein that improves diets worldwide. Cowpea has been used as fortification agents in some traditional corn foods in developing countries such as Colombia. The work aimed to evaluate the physicochemical properties of extruded mixtures of corn and cowpea flours to assess the use of these mixes as vegetable protein ingredients. Corn flour was mixed with 15, 30, and 50% of cowpea flour and extruded for this proposal. After extrusion, mixtures were ground to produce a powder. Techno-functional properties of powders as water content, hygroscopicity, water absorption, fat absorption, water solubility index, swelling index, bulk density, Hausner ratio, Carr index, and porosity were evaluated in the mixtures, extrudates, and obtained powders to assess the effect of the addition of cowpea on these properties. Results showed that processing powder obtained by extrusion and drying could be used as a powder to regenerate with water as a source of protein. Moreover, storing processing samples in sections (pellet format) is convenient to avoid wetting since this format is less hygroscopic and the same mass occupies less storage volume than powders.

## 1. Introduction

The mixture of cereals and legumes is well-known to improve the nutritional value of foods, highlighting their amino acidic profile [1]. These combinations can solve malnutrition problems in developing countries [2]. For example, in Colombia, traditional beverages such as chucula contain a mix of cereals and legumes generally consumed by farmers and habitats in the mountains of small peasant communities [3]. These traditional beverages are also considered as an excellent energy source for their carbohydrate content. This kind of beverage is a mixture of powders dissolved in hot water or milk, with many problems related to its final solubility [3]. One of the legumes used in this beverage is cowpea. Cowpea (*Vigna unguilata*) is a legume widely cultivated in Colombia with great cultural and economic importance, low price, accessibility, and is a source of starch and protein [4]. The proximate composition of cowpea flour is 25% protein, 51% starch, 11% fiber, 3% ash, and 1.5% fat and minerals [5]. Furthermore, cowpea can be considered as a gluten-free food. Another component typically used in this kind of beverage is corn (*Zea mays*). Corn is one of the cereals used more in Latin America as part of native diets as a source of starch and protein. The composition of corn flour is 66% carbohydrates, 8.3% protein, 3% fat, and 9.5% fiber [6]. Despite this ingredient’s importance and nutritional value, some things related to its use could be improved. For example, the nutritional value of cowpeas can be reduced by the presence of anti-nutrients such as trypsin inhibitors, phytates, and tannins [2,7,8,9]. These compounds can reduce the bioavailability of minerals such as calcium, magnesium, iron, zinc, and copper through the formation of insoluble or very poorly dissociated complexes [10]. Another aspect to consider in evaluating the nutritional value of nutrients is their digestibility. The protein digestibility in corn ranges from 85% to 90% [6,11], and in the case of cowpea, it is around 80% [12]. It is well-known that some low-moisture processing techniques such as extrusion can increase digestibility and significantly reduce anti-nutritional factors [13]. Extrusion cooking implies a combination of pressure, high temperature, and shear forces. This process combines unit operations such as cooking, mixing, shearing, and forming [14]. Because of this process, many changes in food can affect its techno-functional and organoleptic properties. Between these changes can be mentioned the gelation of starches, denaturation, or reorientation of proteins, fat melting, or expansion of the food structure [15]. As previously mentioned, there have been many studies on the effect of the extrusion process on the physicochemical properties of starch and proteins [14,16,17,18,19,20]. However, few works have used extrusion to obtain cereal and legume powders as an ingredient. For example, Wang et al. [21] used the extrusion treatment to improve the instant properties of kudzu powders. Szczygiel et al. [22] evaluated the acceptance of navy bean (*Phaseolus vulgaris*) powders prepared by extrusion as a process to eliminate flatulence-causing oligosaccharides and anti-nutrients. Obilana et al. [23] assessed the effects of extrusion on the physical and functional properties of the pearl millet-based instant beverage powder. Diez-Sánchez et al. [24] used extruded flour with blackcurrant pomace as a techno-functional ingredient to obtain a hyperglycemic effect of pre-gelatinized starch in muffins. However, the effect of the extrusion on the techno-functional properties of corn-cowpea extruded powders has never been studied. Therefore, this work aimed to evaluate the effect of the extrusion of mixtures of corn and cowpea on the techno-functional properties of the obtained powders. These properties can be used in the development of different kinds of foods as beverages, plant-based meat replacers, or gluten-free products.

## 2. Materials and Methods

### 2.1. Raw Materials

Cowpea (*Vigna unguilata*) was purchased from a local market (Valencia, Spain). Corn grits were supplied by Maicerías Españolas S.L. (Valencia, Spain).

### 2.2. Formulations and Extrusion Processing

Cowpea was ground in a Thermomix (TM 21, Vorwerk, Valencia, Spain) for 1 min at 5200 rpm. The obtained powder was named Vu. The protein content of this powder was 22 ± 1%.

Corn grits were blended with water in a ratio of 9:1 to obtain the control sample (C). C was mixed with 15, 30, and 50% of cowpea (Vu) to produce the extrusion mixtures (M). CM, 15VuM, 30VuM, and 50VuM were the mixtures used to feed the extruder (Figure 1). Extrusion was carried out using a single-screw laboratory extruder (Kompakt extruder KE 19/25; Brabender, Duisburg, Germany) with a 19 mm diameter barrel and a length:diameter ratio of 25:1, operating with a screw of a 2:1 compression ratio, a constant feeding speed of 18 rpm, feed rate of 3.9 kg/h, and the use of a nozzle 4 mm in diameter. The screw was rotated constantly at 120 rpm, and barrel section (four) temperatures from the feeder to the nozzle were set to 25, 55, 110, and 110 °C, respectively. Motor torque, barrel temperatures, screw speed, and melt pressure were monitored using the Extruder Winext software (Brabender). The pressure measured on the extruder head during the process ranged between 80 and 120 bar. After extrusion, samples were dried at 40 °C for 24 h to reduce the water content of the pellet sections (S). These samples were CS, 15VuS, 30VuS, and 50VuS (Figure 1). Finally, dried pellet sections were ground in a Thermomix (TM 21, Vorwerk, Valencia, Spain) for 1 min at 5200 rpm to produce powder (P) from samples, named CP, 15VuP, 30VuP, and 50VuP (Figure 1).

### 2.3. Determinations

#### 2.3.1. Water Content and Hygroscopicity

Water content (x_w_), expressed as g water/100 g sample, was determined by vacuum oven drying at 105 °C until constant weight [25] for mixtures (M) and final powder (P) with and without cowpea (Vu). Samples were analyzed in triplicate. Water loss because of the extrusion and drying was calculated as the difference between the mixtures and water content of the powders.

Hygroscopicity (Hy) was evaluated in mixtures (M), dried pellet sections (S), and final powder (P). Approximately, 0.5 g of each sample was placed in a Petri dish at 25 °C in a desiccator conditioned at 81% relative humidity with Na_2_SO_4_ saturated solution. After 2, 5, and 7 days, the samples were weighed. The hygroscopicity (Hy) was expressed as g of water gained per 100 g of dry solids [26].

#### 2.3.2. Water and Oil Absorption Index, Water Solubility Index, and Swelling Index

The analyses in this section were performed only on the products that had a powdery appearance (M and P) due to the protocol restrictions. The water solubility index (WSI) and water absorption index (WAI) were determined by the method of Singh and Smith [27]. A 2.5 g sample (M or P) was dispersed in 25 g of distilled water. After stirring for 30 min, the dispersions were rinsed into tared 50 mL centrifuge tubes, made up to 32.5 g, and centrifuged at 3000× *g* for 10 min. After centrifugation, the sediment was weighed, and the supernatant was decanted for the dissolved solid content determination. WAI and WSI were calculated according to Equations (1) and (2).
(1)$$\mathrm{WAI}\;=\frac{\mathrm{weight}\;\mathrm{of}\;\mathrm{sediment}\;}{\mathrm{weight}\;\mathrm{of}\;\mathrm{dry}\;\mathrm{solids}\;}$$
(2)$$\mathrm{WSI}\;\left(\%\right)=\left(\frac{\mathrm{weight}\;\mathrm{of}\;\mathrm{dissolved}\;\mathrm{solids}\;\mathrm{in}\;\mathrm{supernatant}\;}{\mathrm{weight}\;\mathrm{of}\;\mathrm{dry}\;\mathrm{solids}}\right)\times 100$$

Following the method reported by Navarro-González, García-Valverde, García-Alonso, and Periago [28] with little modifications [29], the oil adsorption index (FAI) was determined. For this, samples of 4 g were placed in a centrifuge tube with 24 g of sunflower oil. The tubes were stirred (3000 rpm) for 30 s every 5 min until 30 min. After stirring, samples were centrifuged at 1600× *g* for 25 min, and free oil was decanted. FAI was expressed as g oil/g sample.

For the measurement of the swelling index (SWE), the bed volume technique was used. In brief, samples were weighed (5 g) and transferred to a graduated test tube, and 50 mL of distilled water was added. Test tubes were maintained for 18 h at ambient temperature. The bed volume was measured and expressed as mm of swollen sample per g of the dry initial sample [30].

#### 2.3.3. Bulk Density, Hausner Ratio, Carr Index, and Porosity

For the powders, to determine the bulk density (ρ_b_) in a 10 mL graduated test tube, approximately 2 g of powder was placed. The occupied volume was recorded. ρ_b_ was calculated by dividing the powder mass by the occupied volume and was expressed as g/L. To determine the tap density (ρ_T_), a graduated test tube with 2 g of powder was mechanically tapped, and the volume was recorded until it reached a constant volume. ρ_T_ was calculated by dividing the powder mass by the occupied volume after tapping and was expressed as g/L. For pellet sections, ρ_b_ determination, measurements of the height and diameter of cylinders were taken 15 times with an electronic Vernier caliper (Comecta S.A., Abrera, Spain), and the pellet sections were weighed with a precision scale (±0.001 g) (Mettler Toledo, Greifensee, Switzerland). The samples’ true density (ρ) was established by a helium pycnometer (AccPyc 1330, Micromeritics, Norcross, GA, USA). From these determinations, the Hausner ratio (HR), which is correlated to the flowability of a powder [31], was calculated by Equation (3), and the Carr index (CI), which represents the compressibility of a powder [32], was calculated by Equation (4).
(3)$$\mathrm{HR}=\frac{\rho_{T}}{\rho_{b}}$$
(4)$$\mathrm{CI}=100\times \frac{\rho_{T}-\rho_{b}\;}{\rho_{T}}$$
where HR is the Hausner ratio; CI is the Carr index (%); ρ_b_ is the bulk density (g/L); and ρ_T_ is the tap density (g/L).

Moreover, porosity (ε), the percentage of air volume related to the total volume, was calculated according to Equation (5). For the powdery samples, M and P, this parameter was described by Igual et al. [33], and in the case of the pieces, S, according to Igual et al. [34].
(5)$$\varepsilon =\frac{\left(\rho -\rho_{b}\right)}{\rho }$$
where ε is the porosity; ρ_b_ is the bulk density (g/L); and ρ is the true density (g/L).

#### 2.3.4. Color

CIE*L*a*b* color coordinates were measured considering a standard light source D65 and a standard observer 10° (Minolta spectrophotometer CM-3600d, Japan). Measurements were taken nine times on the mixtures (M), the section of the pellet (S), and powder (P) samples. Previously, samples were measured on white and black backgrounds to consider translucency. In any case, the samples were not translucent. The total color differences between the section of pellet and mixtures or between the final powders to mixtures (ΔE_1_) were calculated, which means that the reference was the correspondent mixture before extrusion. The total color difference (ΔE_2_) was calculated to evaluate the color changes by Vu effect in the mixtures, sections of pellets, and final powders. In this case, the reference was cowpea flour (Vu).

Color coordinates were also measured in the mixtures and powder samples after 18 h in contact with water inside the calibrated cylinder, as described for SWE in Section 2.3.1.

### 2.4. Statistical Analysis

Analysis of variance (ANOVA) was applied with a confidence level of 95% (*p* < 0.05) to evaluate the differences among samples. Additionally, a correlation analysis was conducted among the studied parameters of samples with a 95% significance level. The software Statgraphics Centurion XVII, version 17.2.04 (Statgraphics Technologies, Inc., The Plains, VA, USA) was used to conduct this analysis.

## 3. Results and Discussion

### 3.1. Physicochemical Characteristics of Extrudates

Figure 2 shows the x_w_ of the mixtures and powder obtained after the extrusion and drying process and W_L_ during the extrusion and drying process. The Vu incorporation to the corn grits to feed the extruder significantly decreased (*p* < 0.05) the water content of mixtures in all cases, especially at 50 Vu. W_L_ due to the extrusion and drying process was significantly higher at 15 Vu. In the end, the powder x_w_ was significantly higher in the control than the rest of the samples enriched with Vu. Vu enrichment boosted water loss after the process (extrusion + drying). Total water loss in samples with Vu ranged from 0.125 to 0.149 g_w_/g_db_, whereas the control samples presented 0.066 g_w_/g_db_.

WAI, WSI, SWE, and FAI of the mixtures and powder are shown in Figure 3. The WAI and WSI indices show how samples interact with water [35]. WAI refers to the water absorbed by the samples when immersed in water [36]. WSI refers to the water-solubilized components released during extrusion [37]. Final powder samples (P) presented significantly (*p* < 0.05) higher WAI than the mixtures (M). The WAI parameter shows the capability of a material to absorb water, and is affected by the magnitude of the molecular interaction within the starch structure (amorphous and crystalline). Extrusion implies a gelatinization of starch, and then its crystalline structure is modified due to the breaking of inter- and intra-molecular hydrogen bonds. Therefore, hydroxyl groups are more exposed to form hydrogen bonds with water. As a result, water molecules can diffuse into the amorphous region of extruded starch more easily than in native starch [38]. Materials with high WAI tend to be easily dispersed in water. Hence, the WAI of extruded starch of P was higher than that of the native starch of M, as shown in other studies [39,40,41,42,43,44]. There was no significant (*p* > 0.05) effect of the Vu addition in P; however, this effect was significant (*p* < 0.05) in M. According to other studies carried out with mixtures of starch and vegetable protein [45], there are interactions between both components that increase the capacity for absorption and retention of water. Extrusion + drying reduced the number of soluble components released as indicated by WSI values, thus reducing the molecular degradation in samples. Other studies observed that the WSI of extruded starch is much higher than that of native starch [39,40,41,42,43,44,46]. However, the feed mixtures to the extruder contain water, as indicated in Section 2.2, which acts as a plasticizer and probably reduces shearing and starch degradation during extrusion. Therefore, high feed moisture results in higher WAI and lower WSI [42,46]. The WSI of P samples did not show significant (*p* > 0.05) differences for %Vu. In the case of sample M, the WSI values of the mixtures with 50% Vu were significantly (*p* < 0.05) lower than the rest.

Extrusion + drying significantly increased (*p* < 0.05) the SWE of mixtures with or without Vu (Figure 3). In M, there was a significant effect of Vu % (*p* < 0.05), with higher Vu % in samples with higher SWE. However, in P, only the sample with 50% Vu showed significant (*p* < 0.05) higher values of SWE. The behavior of the SWE is similar to the WAI both because of the processing and the addition of Vu. Gelatinization produces significant structural changes that destroy the light packing of the polymers and allow their release to interact with water [38]. Therefore, P showed more significant swelling when in contact with water than M. If M and P were used as ingredients for the preparation of a food product, P would introduce more water into the network than M. This would give the future product more consistency. At a dietary level, it would provide satiety to the consumer and reduce caloric intake.

The FAI of samples is shown in Figure 3. P presented lower values of FAI than M, but only in samples with 15% Vu and 50% Vu were the FAI differences significant (*p* < 0.05). In both M and P, when Vu % increased, FAI decreased. The incorporation of protein into the mixtures probably caused a lower oil absorption. If M and P were used as an ingredient for the preparation of a food product, a lower FAI would guarantee that if the product is fried or in contact with oil, P will capture less oil than M. In addition, either of the two with Vu would also capture less oil than without Vu.

Figure 4 shows the mean values and standard deviation of ρ_b_, ε, HR, and CI. In the range of samples enriched between 0 and 30% Vu, the samples with the highest ρ_b_ were S, followed by P and M. However, when the mixtures contained 50% Vu in their formulation, the densest was P. The opposite trend was observed in ε, with the most porous being samples M. Although the final use of the products studied in this work was M or P, the S format could be very suitable since, being denser and less porous, it occupies less space and would facilitate transport and storage. In powdered products, porosity plays a key role in the agglomerate strength of dried foods [47]. Furthermore, a greater porosity (and lower bulk density) corresponds to a greater air volume distributed among particles where the water inlet could be more accessible [47,48]. A larger particle size conditions the greater air volume among particles since the particles, when they settle, leave air spaces of a greater volume than if the particles are small. A smaller particle size allows for a better organization of particles. It leaves smaller spaces among particles and, therefore, less porosity, as can be observed for P compared with M. Figure 4 shows HR and CI. HR is an index correlated to the flowability of a powder. The difference range for HR in defining the flowability is free flowing powder (1.0 < HR < 1.1), medium flowing powder (1.1 < HR < 1.25), difficult flowing powder (1.25 < HR < 1.4), and tough flowing powder (HR > 1.4) [49]. According to this ranking, P presents a difficult flowing powder, but in the case of 50VuP, it was close to the medium flowing powder. However, M showed tough flowing powder for all formulations. CI represents the compressibility of a powder. According to Carr [32], excellent flowability can be expected if the CI is within 5 to 15%, and if the value of CI is above 25%, it indicates poor flowability. The CI of M samples ranged from 53 to 68%, so they presented poor flowability. However, the values of CI for 30VuP and 50VuP were 25 and 21%, respectively, showing an intermediate flowability. There were significant (*p* < 0.05) differences between M and P when comparing the HR and CI values. The use of Vu in the formulations provoked a significant (*p* < 0.05) decrease in HR and CI, improving the flowability properties of the samples.

Figure 5 shows the evolution of the hygroscopicity of samples along 7 d. Hygroscopicity can be defined as the capacity of a powder to absorb water from the environment. This property can determine the stability of products during storage. Samples with lower hygroscopicity are desirable for handling and packaging [50]. After extrusion + drying, S and P showed significantly higher Hy values than M. Comparing S and P, S was less hygroscopic than P, so after the studied process (extrusion + drying), it was convenient to store in sections (pellet format) since S presented less hygroscopicity than P. Once its use is required, it can be crushed, and a powder will be obtained. In addition, S is denser than P, as indicated above, and therefore, the same mass occupies less volume in storage. In S and P, there was a significant (*p* < 0.05) and increasing effect with an increasing concentration of Vu. However, only 50VuM showed significant differences in Hy in the M samples.

Pearson correlations (Table 1) were performed to explore the relationships among the studied properties of the samples. x_w_ maintained a close relationship with all the properties studied. Significant (*p* < 0.05) and positive correlations were observed with WSI, FAI, HR, CI, and ε, and negative with WAI, SWE, Hy, and ρ_b_. In all cases, the correlation coefficients were greater than 0.94, except for FAI, which was 0.78.

The flowability of the powder decreased (increase in HR and CI) when the water content increased in the same way as in other studies of barberry powders [51]. Moreover, HR and CI were significant (*p* < 0.05) and positively correlated with ε (negatively with ρ_b_), so highly porous powders will be difficult to flow, probably due to the larger particle size or higher water content. In Table 1, a significant positive Pearson’s correlation between ε and WSI could also be observed. A greater porosity and lower bulk density corresponded to a more soluble product [47,48]. Hy presented a significant Pearson’s correlation related to ρ_b_ and ε (*p* < 0.05). The higher the Hy, the higher the ρ_b_ and the lower the ε, with 0.9050 and −0.9043 correlation values, respectively, as was observed in other studies with beetroot powder [50].

The color of the mixtures (M), extruded pellet section (S), and final powders (P) are shown in Figure 6. The difference between S and M or P in L*, a*, and b* stands out. As shown in Figure 1, M and P were more luminous than S, following the L* values in Figure 6. No trend was observed as defined by the addition of Vu in L*. Overall, S showed higher values of a* and lower values of b* compared to M and P. These differences are due to the matrix of the products. The total color differences of S and P with M (Figure 6) showed that the more compact matrix of S presented greater total color differences for M at all concentrations of Vu. ΔE2 presented significant differences due to Vu %. The addition of Vu significantly increased (*p* < 0.05) the ΔE2 values in M, however, in the P samples, the opposite trend was observed (significant decrease, *p* < 0.05). In the case of S, significant differences in ΔE2 were only observed when the sample contained 50% Vu in its composition.

### 3.2. Color of Hydrated Samples

Considering the results of the physical properties measured in this work and the current trends in the use of vegetable protein, the samples hydrated for 18 h were the object of study. In this way, the color coordinates and appearance of samples hydrated for 18 h are shown in Table 2 and Figure 7, respectively. For each %Vu, there were significant (*p* < 0.05) differences in the color coordinates between M and P, showing higher values M. At the four concentrations of Vu studied, the ΔE1 was greater than three and therefore perceptible to the human eye [33]. Within the M samples, 50VuM showed significant differences (*p* < 0.05) in L*, a*, and b*, more marked than the rest. In the P samples, while L* and b* decreased with increasing Vu %, a* was stable. ΔE2 for both M and P presented significant (*p* < 0.05) differences due to the % Vu. The appearance of P samples (Figure 7) was homogeneous and compact. These samples can be used as a vegetable matrix to substitute proteins of animal origin. However, the M samples present an appearance of loose particles and a non-homogeneous matrix. This appearance change is due to processing (extrusion + drying), since there is no effect of Vu %.

For each parameter, the same small letter indicates homogeneous groups established by ANOVA (*p* < 0.05) by comparing the Vu percentages (0, 15, 30, and 50) in M or P. For each Vu percentage (0, 15, 30, and 50) and parameter, the same capital letter indicates homogeneous groups established by ANOVA (*p* < 0.05) by comparing M and P.

## 4. Conclusions

Extruded powders from corn and cowpea could be obtained to provide new protein-complement ingredients. The techno-functional proprieties studied showed that processing powder obtained by extrusion and drying could be used as a powder to regenerate with water as a source of protein. Extrusion and drying reduce the moisture in the samples, increasing their stability during storage. On the other hand, the extrusion and drying of mixtures of corn and cowpea increase the WAI of obtained powders, improving their dispersibility in water. No significant effect of the addition of cowpea was found on WAI. Extrusion plus drying reduced the number of soluble components released, as indicated by the WSI values, thus reducing the molecular degradation in the samples. Moreover, it is convenient to store processing samples in sections (pellet format) to avoid wetting, since this format is less hygroscopic and the same mass occupies less storage volume than powders. The cowpea addition caused significant color samples in the mixtures and powders. In the powder samples, while L* and b* decreased with increasing cowpea, a* was stable. The appearance of powder samples resulted in more homogeneity and compactness than the mixtures. It would be advisable to continue studying the processed samples so that further progress can be made toward understanding each component’s role in imparting functional characteristics to foods.

## Figures and Tables

**Figure 1 foods-12-00298-f001:**
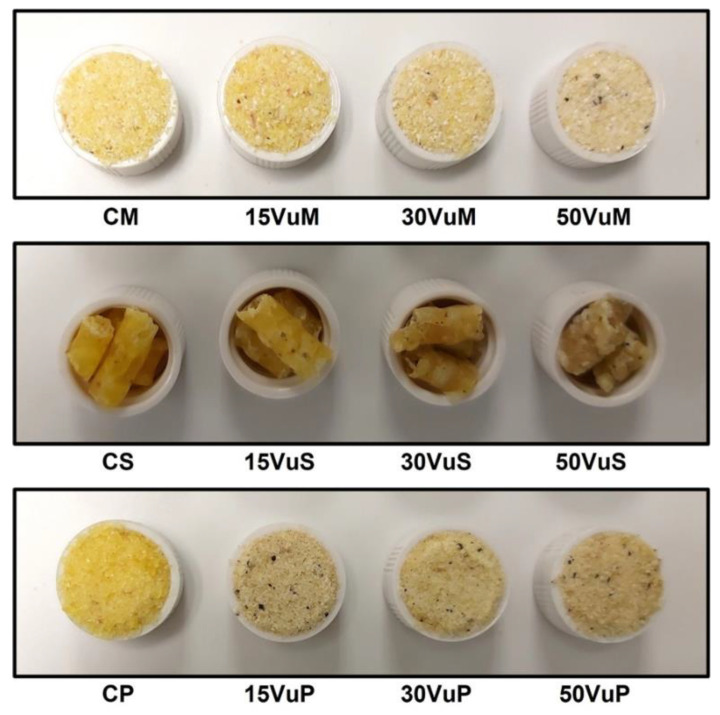
The studied samples. Mixtures (M), pellet sections (S), and powder (P) with different concentrations (15, 30, and 50%) of *Vigna unguilata* (Vu) and the respective controls (CM, CS, and CP).

**Figure 2 foods-12-00298-f002:**
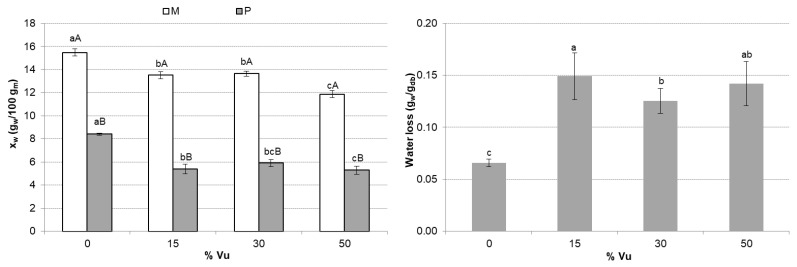
Mean values and standard deviation of moisture (x_w_) of the mixtures (M) and powder (P), and the total water loss (W_L_) of the process. For each parameter, the same small letter indicates homogeneous groups established by ANOVA (*p* < 0.05) by comparing the Vu percentages (0, 15, 30, and 50). For each Vu percentage (0, 15, 30, and 50) and parameter, the same capital letter indicates homogeneous groups established by ANOVA (*p* < 0.05) by comparing M and P.

**Figure 3 foods-12-00298-f003:**
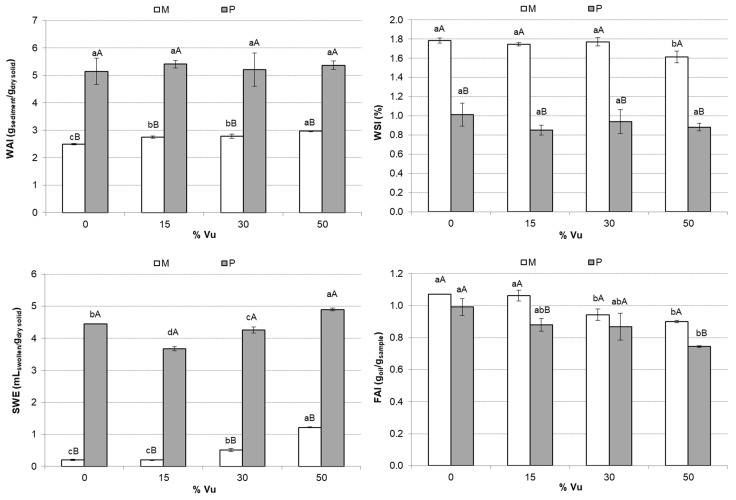
Mean values and standard deviation of the water absorption index (WAI), water solubility index (WSI), swelling index (SWE), and fat absorption index (FAI) of the mixtures (M) and powder (P). For each parameter, the same small letter indicates homogeneous groups established by ANOVA (*p* < 0.05) by comparing the Vu percentages (0, 15, 30, and 50) in M or P. For each Vu percentage (0, 15, 30, and 50) and parameter, the same capital letter indicates homogeneous groups established by ANOVA (*p* < 0.05) comparing M and P.

**Figure 4 foods-12-00298-f004:**
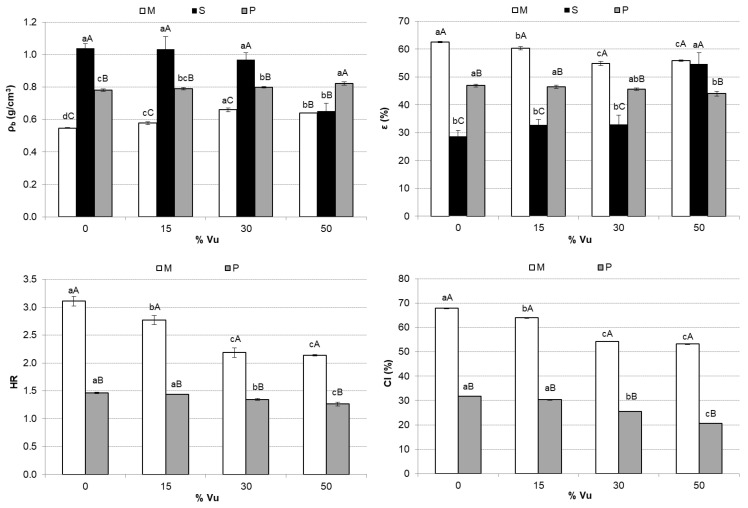
Mean values and standard deviation of the bulk density (ρ_b_), porosity (ε), Hausner ratio (HR), and Carr index (CI) of the mixtures (M), the section of the pellet (S), and powder (P) in each case. For each parameter, the same small letter indicates homogeneous groups established by ANOVA (*p* < 0.05) by comparing the Vu percentages (0, 15, 30, and 50) in M, S, or P. For each Vu percentage (0, 15, 30, and 50) and parameter, the same capital letter indicates homogeneous groups established by ANOVA (*p* < 0.05) by comparing M, S, and P.

**Figure 5 foods-12-00298-f005:**
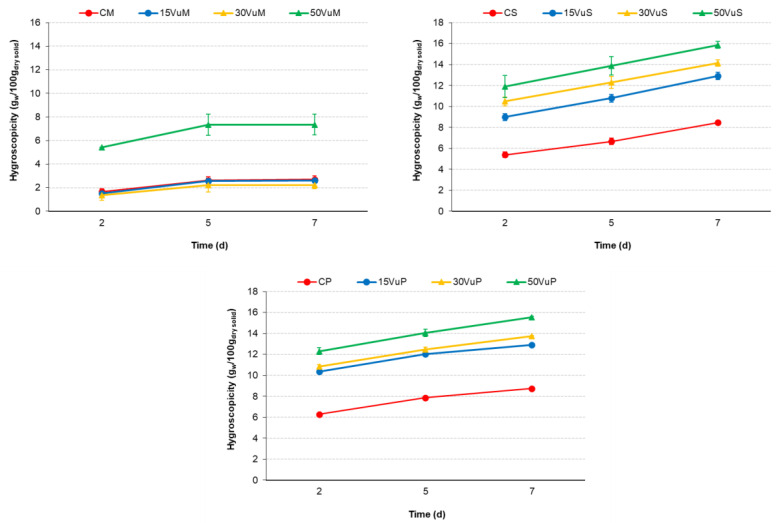
Hygroscopicity of each powder sample along with assay time. Mixtures (M): CM, 15VuM, 30VuM, and 50VuM; Section of pellet (S): CS, 15VuS, 30VuS, and 50VuS; Powder (P): CP, 15VuP, 30VuP, and 50VuP.

**Figure 6 foods-12-00298-f006:**
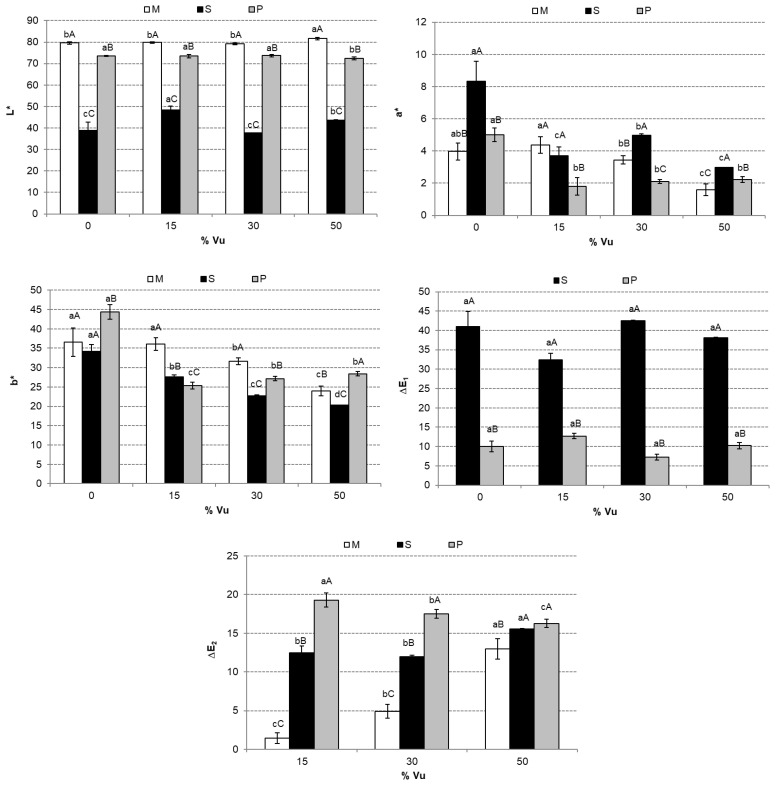
Mean values and standard deviation of color coordinates (L*, a*, and b*) and the total color differences (ΔE_1_ and ΔE_2_) of the mixtures (M), the section of the pellet (S), and powder (P). For each parameter, the same small letter indicates homogeneous groups established by ANOVA (*p* < 0.05) by comparing the Vu percentages (0, 15, 30, and 50) in M, S, or P. For each Vu percentage (0, 15, 30, and 50) and parameter, the same capital letter indicates homogeneous groups established by ANOVA (*p* < 0.05) by comparing M, S, and P.

**Figure 7 foods-12-00298-f007:**
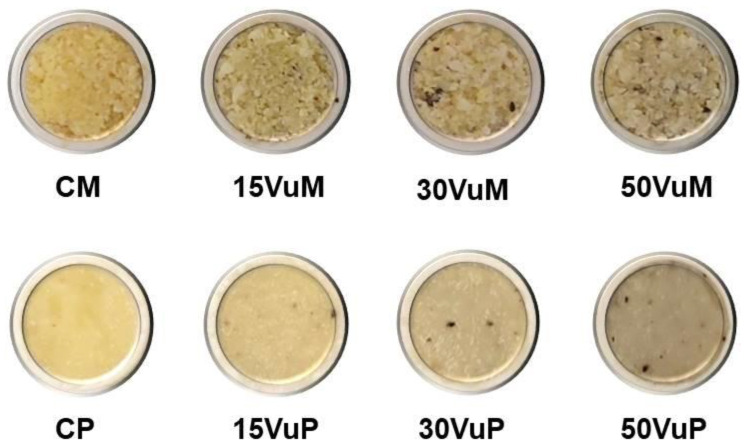
The appearance of the mixtures (M) and powder (P) with different concentrations (15, 30, and 50%) of *Vigna unguilata* (Vu) and the respective controls (CM and CP), hydrated for 18 h according to the SWE methodology.

**Table 1 foods-12-00298-t001:** Pearson correlation coefficients among the studied physicochemical parameters of powders. Water content (x_w_), water absorption (WAI), water solubility (WSI) index, fat absorption index (FAI), swelling index (SWE), hygroscopicity (Hy), bulk density (ρ_b_), Hausner ratio (HR), Carr index (CI), and porosity (ε).

	WAI	WSI	FAI	SWE	Hy	ρ_b_	HR	CI	ε
**x_w_**	−0.9725 *	0.9794 *	0.7819 *	−0.9464 *	−0.9679 *	−0.9606 *	0.9433 *	0.9720 *	0.9574 *
WAI		−0.9963 *	−0.6667 *	0.9791 *	0.9783 *	0.9653 *	−0.9296 *	−0.9695 *	−0.9584 *
WSI			0.6847	−0.9746 *	−0.9413 *	−0.9540 *	0.9193 *	0.9641 *	0.9470 *
FAI				−0.6987	−0.8388 *	−0.7803 *	0.8046 *	0.8023 *	0.7928 *
SWE					0.9212 *	−0.9627 *	−0.9347 *	−0.9773 *	−0.9604 *
Hy						0.9050 *	−0.8892 *	−0.9398 *	−0.9043 *
ρ_b_							−0.9878 *	−0.9936 *	−0.9994 *
HR								0.9776 *	0.9914 *
CI									0.9934 *

* Correlation was significant at 0.05. All data represent the mean of three determinations.

**Table 2 foods-12-00298-t002:** Mean values (and standard deviations) of color coordinates (L*, a* and b*) and total color differences (ΔE_1_ and ΔE_2_) of the mixtures (M) and powder (P) with different concentrations (15, 30, and 50%) of *Vigna unguilata* (Vu) and the respective controls (CM and CP) hydrated for 18 h according to SWE methodology.

Sample	L*	a*	b*	ΔE_1_	ΔE_2_
CM	65.04 (0.09) ^bA^	2.42 (0.12) ^aA^	30.5 (0.4) ^aA^		
15VuM	63.4 (0.7) ^cA^	0.4 (0.4) ^bA^	23.96 (1.03) ^bA^		7.1 (0.9) ^cA^
30VuM	65.9 (0.9) ^bA^	0.57 (0.18) ^bA^	20.61 (1.02) ^cA^		10.2 (0.9) ^bA^
50VuM	68.0 (0.4) aA	−0.13 (0.14) ^cA^	16.9 (0.7) ^dA^		14.2 (0.6) ^aA^
CP	63.83 (0.18) ^aB^	−1.28 (0.02) ^aB^	18.0 (0.2) ^aB^	13.1 (0.2) ^b^	
15VuP	59.4 (0.5) ^bB^	−1.13 (0.16) ^aB^	15.8 (0.03) ^bB^	9.2 (0.2) ^d^	5.0 (0.3^) cB^
30VuP	59.8 (0.8) ^bB^	−1.31 (0.08) ^aB^	12.5 (0.7) ^cB^	10.34 (1.12) ^c^	6.84 (1.02) ^bB^
50VuP	55.3 (0.5) ^cB^	−1.2 (0.3) ^aB^	10.2 (0.4) ^dB^	14.4 (0.3) ^a^	11.6 (0.2) ^aB^

For each parameter, the same small letter indicates homogeneous groups established by ANOVA (*p* < 0.05) by comparing the Vu percentages (0, 15, 30, and 50) in M or P. For each Vu percentage (0, 15, 30, and 50) and parameter, the same capital letter indicates homogeneous groups established by ANOVA (*p* < 0.05) by comparing M and P.

## Data Availability

The data are avaliable from the corresponding author.

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
