# Peer review of "Techno-Functional Properties of Corn Flour with Cowpea (Vigna unguilata) Powders Obtained by Extrusion"

_foods, 2023, doi:10.3390/foods12020298_

Round 1

Reviewer 1 Report

Introduction is very general. Introduction must be about the focus of the article. So, introduction must be related with corn- Leguminosae flours and physicochemical properties.

Title: if author dry his material in a special way then it is necessary incorporate in title; if drying is in a normal way it is not necessary, ie, if dry is a variable in the design experiments then is necessary mention in the title.

Line 65-67: It is not clear which was the final moisture of the blends. Please add which was the final moisture before feed it at extruder.

Line 70: compression ratio is fixed. Which was the compression ratio of the screw?

Line 72: How many barrel sections has the extruder? Add in the text.

Line 96: Change fat by oil. Fat has another meaning than oil.

Line 136: Add the meaning of each letter or variable in the bottom of each equation.

Ro (b) is nit defined in the text. Add at the bottom of the equation

Line 144: it is not clear which was the reference for calculate delta E.

Line 166 . Values in text not match with value in figure 2.

Line 225- 224.  Porosity in food has different meaning. Porosity is not the air space in a grain milled. Porosity is the inlet air on sample. When flour is heated in extrusion, flour is expanded by temperature and pressure generating or increasing its porosity.  Author must rewrite his results.

Why M samples has the maximum porosity? It is possible? Endosperm is porous?

Author is confused about optical properties.  In foods, optical properties means UV- Vis analysis. In UV-Vis spectroscopy color is inherent in the measuring.

Conclusion: Author did not conclude anything about his results. Only mention about pellets. If the focus of the article are the pellets propeeties then author must rewrite title and article.

Author Response

Response to Reviewer 1 Comments

Manuscript ID: foods-2095101

Reviwer 1

Comments and Suggestions for Authors

Introduction is very general. Introduction must be about the focus of the article. So, introduction must be related with corn- Leguminosae flours and physicochemical properties.

Response: Introduction has been completed according to reviewer suggestion.

 Title: if author dry his material in a special way then it is necessary incorporate in title; if drying is in a normal way it is not necessary, ie, if dry is a variable in the design experiments then is necessary mention in the title.

Response: Title has been modified as “Physicochemical properties of corn flour with cowpea (Vigna unguilata) powders obtained by extrusion.”

 Line 65-67: It is not clear which was the final moisture of the blends. Please add which was the final moisture before feed it at extruder.

Response: In Figure 2, mean values and standard deviation of mixtures/blends (M) moisture are represented.

Line 70: compression ratio is fixed. Which was the compression ratio of the screw?

Response: Authors mentioned the characteristics of the screw, the compression ratio of this screw is 2:1. The sentence has been rewritten as: “…with a barrel diameter of 19 mm and a length:diameter ratio of 25:1, operating with a screw of 2:1 compression ratio, a constant feeding speed of 18 rpm, and using a nozzle of 4 mm of diameter.”

Line 72: How many barrel sections has the extruder? Add in the text.

Response: The sentence has been rewritten as follows: “The screw was rotated constantly at 120 rpm and temperatures of barrel sections (four) from feeder to the nozzle were set to 25, 55, 110, and 110 °C respectively.

Line 96: Change fat by oil. Fat has another meaning than oil.

Response: Change was made

Line 136: Add the meaning of each letter or variable in the bottom of each equation.

Response: The meaning of each letter or variable was added

Ro (b) is nit defined in the text. Add at the bottom of the equation

Response: The meaning of each letter or variable was added

Line 144: it is not clear which was the reference for calculate delta E.

Response: This point has been clarified in the text.

Line 166 . Values in text not match with value in figure 2.

Response: Values have been corrected.

Line 225- 224.  Porosity in food has different meaning. Porosity is not the air space in a grain milled. Porosity is the inlet air on sample. When flour is heated in extrusion, flour is expanded by temperature and pressure generating or increasing its porosity.  Author must rewrite his results.

Response: In the manuscript, the porosity of both M and P, powdery products, and S, corresponding to the pellets obtained by extrusion, have been determined. As suggested by the reviewer, in the case of S, the porosity of a structure is the volume of space within that structure. However, in the case of powdered products, the porosity refers to the volume of space between particles in a known volume of powder. To further clarify this parameter, explanations for each format have been introduced in the manuscript in the materials and methods section, along with the reference used for the methodology for powdered products and parts.

The final text is: “Moreover, porosity (ε) that is the percentage of air volume related to the total volume, was calculated according to equation 5. For the powdery samples, M and P, this parameter was described by Igual et al. [15], and in the case of the pieces, S, according to Igual et al. [16].

Why M samples has the maximum porosity? It is possible? Endosperm is porous?

Response: As can be seen in figure 1 of manuscript M, these are powdery samples, obtained by mixing corn and cowpea.  According to the results obtained (Figure 4), the porosity of the mixtures, and prior to extrusion and drying, is higher than the samples after the treatments studied (P).

Author is confused about optical properties.  In foods, optical properties means UV- Vis analysis. In UV-Vis spectroscopy color is inherent in the measuring.

Response: Thank you for this appreciation as you mentioned, optical properties of foods are those properties which govern how food materials respond to absorption of electromagnetic radiation in the range of optical wavelengths and frequencies. These include visible light and color, but also transmission, reflection and refraction of visible light. Authors changed optical properties by color.

Conclusion: Author did not conclude anything about his results. Only mention about pellets. If the focus of the article are the pellets properties then author must rewrite title and article.

Response: Title and conclusions have been rewritten according reviewer suggestions.

Reviewer 2 Report

line 62-63 - Cowpea was grinded in a Thermomix (TM 21, Vorwerk, Valencia, Spain) for 1 min 62 at 5,200 rpm. Obtained powder was named Vu. The protein content of this powder was 63 22±1%.- really a thermomix kitchen device is used in the laboratory, not a laboratory grinder??????? has the protein also been labeled in Termomix?

We initially overly generalized the extrusion process, which is actually crucial for the research of this work.

in chapter 2.2. it is missing an extremely important parameter like the level of humidity, which plays a key role in the process of starch gelatinization or melting, and thus complexing whites to such an extent that they can be harmful when used in food. It was also forgotten to indicate the parameters of the output nozzle, the narrowing of which during the extrusion of the hot extruded mass causes its sudden expansion and cooling, while forming a porous structure.

line 92-95 -To measure hygro-91 scopicity (Hy), approximately 0.5 g of each extruded sample were placed in a petri dish at 92 25 °C, in an airtight plastic container containing with Na2SO4 saturated solution (81% rel-93 ative humidity). After 2, 5 and 7 days each sample was weighed and the hygroscopicity 94 (Hy) was expressed as g of water gained per 100 g dry solids [7].-  it would be much more correct to define this method as water vapor sorption kinetics.

line 106 - if we write the WAI index - it is expressed in % as WSI

line 114-115 - Samples (4 g) were placed in a centrifuge tube with 24 g of sunflower oil. Con-114 tents were stirred for 30 s every 5 min for 30 min. - were the samples mixed, I guess it was homogenization. In studies of such quantities, precision is necessary - please indicate the revolutions per minute used - of the homogenizer

line 123 - For powders, to determine bulk density... - so far I have not read when the powder was obtained from the porous cylinder-shaped extrudate, what was its grammage, i.e. what mesh size was it sieved through and which fraction was sent for analysis ?

Author Response

Response to Reviewer 2 Comments

Manuscript ID: foods-2095101

Reviwer 2

line 62-63 - Cowpea was grinded in a Thermomix (TM 21, Vorwerk, Valencia, Spain) for 1 min 62 at 5,200 rpm. Obtained powder was named Vu. The protein content of this powder was 63 22±1%.- really a thermomix kitchen device is used in the laboratory, not a laboratory grinder??????? has the protein also been labeled in Termomix?

Response: Authors have used this equipment is several works without problems, and without differences with other laboratory grinders. We use this equipment due its high capacity. Protein was evaluated with an elemental Analyzer CHNS.

We initially overly generalized the extrusion process, which is actually crucial for the research of this work.

Response: In other works, authors study extrusion variables deeply. In this case and based in previous works authors used a previous tested conditions to obtain a pellet with cowpea flour.

in chapter 2.2. it is missing an extremely important parameter like the level of humidity, which plays a key role in the process of starch gelatinization or melting, and thus complexing whites to such an extent that they can be harmful when used in food. It was also forgotten to indicate the parameters of the output nozzle, the narrowing of which during the extrusion of the hot extruded mass causes its sudden expansion and cooling, while forming a porous structure.

Response: In Figure 2, mean values and standard deviation of mixtures (M) moisture are represented. The characteristics of the nozzle are described in the text: “…nozzle of 4 mm of diameter”. Authors don’t know which parameters refer the reviewer.

line 92-95 -To measure hygroscopicity (Hy), approximately 0.5 g of each extruded sample were placed in a petri dish at 25 °C, in an airtight plastic container containing with Na2SO4 saturated solution (81% relative humidity). After 2, 5 and 7 days each sample was weighed and the hygroscopicity 94 (Hy) was expressed as g of water gained per 100 g dry solids [7].-  it would be much more correct to define this method as water vapor sorption kinetics.

Response: Authors follows referenced methods to determine hygroscopicity. In this case is not a kinetic, the method needs to achieve constant weight and for this reason measures at different times were taken.

line 106 - if we write the WAI index - it is expressed in % as WSI

Response: Normally, according to bibliography, water absorption index (WAI) is expressed as grams of sediment by grams of dry solid and water solubility index in %

line 114-115 - Samples (4 g) were placed in a centrifuge tube with 24 g of sunflower oil. Contents were stirred for 30 s every 5 min for 30 min. - were the samples mixed, I guess it was homogenization. In studies of such quantities, precision is necessary - please indicate the revolutions per minute used - of the homogenizer

Response: revolutions per minute used of homogenizer were added in the text.

line 123 - For powders, to determine bulk density... - so far I have not read when the powder was obtained from the porous cylinder-shaped extrudate, what was its grammage, i.e. what mesh size was it sieved through and which fraction was sent for analysis?

Response: In lines 76-80, “After extrusion, samples were dried at 40 °C for 24 h to reduce water content of pellet sections (S). These samples were named CS, 15VuS, 30VuS and 50VuS (Figure 1). Finally, pellet sections were grinded in a Thermomix (TM 21, Vorwerk, Valencia, Spain) for 1 min at 5,200 rpm to produce powder (P) from samples extruded and dried, naming as CP, 15VuP, 30VuP and 50VuP (Figure 1).

Round 2

Reviewer 1 Report

Introduction is very general. Introduction must be about the focus of the article. So, introduction must be related with corn flours and cowpea and what has been published about them in the literature.

So, author can find the contribution and the objective. Objective is the characterization of flours corn-cowpea or it is  the improved nutritional  quality by the addition of cowpea? Or is the effect of extrusion in the techno functional properties? How extrusion gives and product to be used as beverage ingredient? What kind of ingredient? How author decided that application of extruded flour is good for beverage?  

What is the effect of the extrusion in this techno functional properties? Apply statistics to known the correlation between properties, extrusion conditions and concentration of cowpea.

Author need improved quality of figures.

In all figures,  it is necessary set  big letters and numbers because in two columns format figures would be very small.

Add next subtitles:

Line 317:    Color of extruded samples

Line 336:   3.2. Color of hydrated samples

Author measure techno functional properties WAI,WSI,SWE, OAC. ISA, color, hygroscopicity, water loss, moisture. Therefore, I recommend rewrite title adding  techno fucntional properties indeed physicochemical properties.

Conclusion: Author did not conclude anything about his results. Only mention about pellets. If the focus of the article are the pellets properties then author must rewrite title and article.

Conclusion must take the experimental results and with those values correlate with the possible  applications.  Author mention that extruded power can be used in beberage. Which technofunctional properties correlate?.

Hygroscopicity: which is the application of a powder with that value of hygroscopicity?. Absorption  water? . In which application can absorption oil  be used?

According with yours results how you can conclude about the application?  Which technofunctional properties  gives you the application of to be an ingredient in a beverage? What about solubility?

Author Response

Response to Reviewer 1 Comments

Manuscript ID: foods-2095101

Reviwer 1

Introduction is very general. Introduction must be about the focus of the article. So, introduction must be related with corn flours and cowpea and what has been published about them in the literature. So, author can find the contribution and the objective. Objective is the characterization of flours corn-cowpea, or it is the improved nutritional quality by the addition of cowpea? Or is the effect of extrusion in the techno functional properties? How extrusion gives and product to be used as beverage ingredient? What kind of ingredient? How author decided that application of extruded flour is good for beverage? 

Introduction has been rewritten according to reviewer suggestions

What is the effect of the extrusion in this techno functional properties? Apply statistics to know the correlation between properties, extrusion conditions and concentration of cowpea.

In this paper correlation with extrusion conditions are not possible to stablish because of only a work condition is evaluated (speed of the screw, temperature conditions in the barrel, feed speed, and nozzle diameter). The effect of cowpea concentration on techno functional properties is analysed by means analysis of variance for each studied properties to evaluate the differences among samples

 Author needs improved quality of figures.

Figures have been improved (size and resolution)

In all figures, it is necessary set big letters and numbers because in two columns format figures would be very small.

Figures have been improved (size and resolution)

Add next subtitles:

Line 317:    Color of extruded samples

Done

Line 336:   3.2. Color of hydrated samples

Done

Author measure techno functional properties WAI, WSI, SWE, OAC. ISA, color, hygroscopicity, water loss, moisture. Therefore, I recommend rewrite title adding techno functional properties indeed physicochemical properties.

Title has been changed es follows: “Techno functional properties of corn flour with cowpea (Vigna unguilata) powders obtained by extrusion”.

Conclusion: Author did not conclude anything about his results. Only mention about pellets. If the focus of the article are the pellets properties, then author must rewrite title and article. Conclusion must take the experimental results and with those values correlate with the possible applications.  Author mention that extruded power can be used in beverage. Which techno-functional properties correlate?

Conclusions have been rewritten

Hygroscopicity: which is the application of a powder with that value of hygroscopicity? Absorption water? In which application can absorption oil be used?

The parameters indicated by the reviewer characterise a powdered product. Hygroscopicity refers to the stability of the powder when subjected to contact with an atmosphere of high relative humidity and to study the water uptake of that product at different exposure times. If a powder is very hygroscopic it will be less stable. Water and oil absorption parameters are very useful to study the reaction of the sample when in contact with water or oil, respectively. In this way, their behaviour can be predicted when the product is used to formulate food and water or oil is required in its formulation, for example to create meat analogues with the products studied in this work. The products shown in this work could also be used as coating for battered products and these parameters are of great interest for example to observe oil uptake in a frying process.

According with yours results how you can conclude about the application?  Which techno-functional properties gives you the application of to be an ingredient in a beverage? What about solubility?

Water solubility index (WSI) indicates the water solubilized components released during extrusion that can cause molecular damage. However, as suggested reviewer, this index is also related to the ability of its components to dissolve in water. In this case, the processes studied (extrusion plus drying) reduce WSI, avoiding molecular degradation and decreasing water solubility. However, by having a higher WAI, the samples absorb more water and if we look at figure 7 we can see that the P samples are a homogeneous mass comparable to a puree. According to the results, the proposed use of the products obtained would be more suitable for purees or ingredients for the formulation of meat analogues. Thank you for your reflective comments.
